# Visualized Lead Selection for Arrhythmia Classification Based on a Lead Activation Heatmap Using Multi-Lead ECGs

**DOI:** 10.3390/bioengineering11060578

**Published:** 2024-06-07

**Authors:** Heng Wang, Tengqun Shen, Shoufen Jiang, Jilin Wang, Yijun Ma, Yatao Zhang

**Affiliations:** 1School of Mechanical, Electrical and Information Engineering, Shandong University, Weihai 264209, China; 202100800620@mail.sdu.edu.cn (H.W.); 202337585@mail.sdu.edu.cn (J.W.); 202337580@mail.sdu.edu.cn (Y.M.); 2Department of Neurology, Weihai Municipal Hospital Affiliated to Shandong University, Weihai 264200, China; lqsdhte1314@163.com; 3School of Business, Shandong University, Weihai 264209, China

**Keywords:** arrhythmia heartbeat, visualized lead selection, multi-lead ECG

## Abstract

Visualizing the decision-making process is a key aspect of research regarding explainable arrhythmia recognition. This study proposed a visualized lead selection method to classify arrhythmia for multi-lead ECG signals. The proposed method has several advantages, as it uses a visualized approach to select effective leads, avoiding redundant leads and invalid information. It also captures the temporal dependencies of ECG signals and the complementary information between leads. The method deployed a lead activation heatmap (LA heatmap) based on a lead-wise network to select the proper 5 leads from 12-lead ECG heartbeats extracted from the public 2018 Chinese Physiological Signal Challenge database (CPSC 2018 DB), which were then fed into a ResBiTime network combining bidirectional long short-term memory (Bi-LSTM) networks and residual connections for a classification task of nine heartbeat categories (i.e., N, AF, I-AVB, RBBB, PAC, PVC, STD, LBBB, and STE). The results indicate an average precision of 93.25%, an average recall of 93.03%, an average F1-score of 0.9313, and that the proposed method can effectively extract additional information from ECG heartbeat data.

## 1. Introduction

Currently, cardiovascular diseases have exceeded tumor diseases in terms of morbidity and mortality [1], and arrhythmia is a significant branch of cardiovascular diseases. The classification of heartbeats in electrocardiograms (ECGs) is crucial for detecting arrhythmias [2]. In fact, deep learning has become the most popular technology for the automatic classification of ECG heartbeats.

Some recent studies have employed deep learning for single-lead ECG signals. Kachuee et al. [3] used a 1-D CNN to identify five distinct heartbeats of arrhythmias by the lead II of ECG signals. Jahmunah et al. [4] trained DenseNet and CNN models for each lead in a 12-lead ECG and utilized Grad-CAM for model output interpretation. The mentioned methods use single-lead ECG signals to analyze ECG signals, which reduce computational complexity so that the learning model can be trained faster. However, single-lead ECG signals provide limited information. In addition, single-lead signals cannot provide useful information if contaminated or improperly connected. As a result, some researchers began to focus on multi-lead signals in order to obtain more robust information. Jiang et al. [5] fed 12-lead ECG signals into a multi-model multi-scale network to classify nine types of arrhythmias. Li et al. [6] transformed 12-lead ECG signals into 2-D forms and created a universal model with detached squeeze-and-excitation modules for automatic classification of normal rhythm and eight cardiac arrhythmias. Qin et al. [7] proposed a neural network architecture based on deformable CNN to learn the time-domain and lead features of 12-lead ECG signals for auto-detection of arrhythmias. Cai et al. [8] designed a 1-D deep densely connected neural network to detect atrial fibrillation (AF) in 10 s, 12-lead ECG signals. Wang et al. [9] designed a multi-scale fusion network that can capture abnormal patterns of diseases by multi-scale feature extraction from 12-lead ECG signals. In most cases, multi-lead ECG signals can provide more stable information than single-lead ECG signals; however, some redundancy and irrelevant information can be contained in multi-lead signals. Actually, the performance of a learning model can be degraded when some of the 12-lead ECG signals are improperly placed or for other reasons, so that the leads provide invalid or incorrect information. Therefore, it is necessary to select the most useful leads from the 12 leads of ECG signals.

Some current studies have selected several leads from the 12 leads of ECG signals for arrhythmia analysis. Cho et al. [10] used limb 6-lead ECG signals—specifically leads I, II, III, aVR, aVL, and aVF—from 12-lead ECGs to develop a deep learning-based artificial intelligence algorithm (DLA) for detecting myocardial infarction (MI). Kumar et al. [11] selected leads II, V2, and V6 from 12-lead ECG signals to design a LSTM network for cardiovascular disease detection. Li et al. [12] used leads II and V1 from 12-lead ECG signals to develop a 31-layer 1-D residual convolutional neural network for arrhythmia classification. However, using single or reduced ECG leads for the automatic and accurate diagnosis of heart diseases still presents a challenge [13,14]. The mentioned methods select several leads from 12 leads relying only on experience or random selection. Thus, the selected leads may not be valid, so that the performance of the learning model is unstable. Additionally, interpretability and visual decision-making have always been urgent issues in automatic diagnosis, as is the selection of useful leads from the 12-lead ECG. Therefore, there is a need to design a visualized and explainable lead selection method for analyzing arrhythmias.

This study proposed a visualized lead selection method based on the lead activation heatmap (LA heatmap) and a ResBiTime network that combines bidirectional long short-term memory (Bi-LSTM) networks with residual connections. The main contributions of this study are as follows:(1)The lead selection method can choose valid leads rather than redundant ones in a visual and explainable way.(2)The proposed LA heatmap enhances the interpretability of deep learning methods in arrhythmias detection.(3)The ResBiTime network merges Bi-LSTM networks with residual connections to obtain not only the temporal features of ECG signals, but also the complementary information among different leads.(4)The proposed method achieved satisfactory results in the 9-class classification of arrhythmias within the CPSC 2018 DB.

The structure of this paper is as follows. Section 2 introduces data and the proposed method. Section 3 provides results. Section 4 is the discussion, and includes the ablation experiment and interpretability explanation. Section 5 presents the conclusion of this study.

## 2. Materials and Methods

### 2.1. Database

The First China Physiological Signal Challenge 2018 database (CPSC2018) [15] was used to validate the proposed method in this study; the database was collected from 3699 male and 3178 female patients from 11 hospitals, and included a total of 6877 12-lead signals with recording times ranging from 6 to 60 s and a sampling rate of 500 Hz. All signals were categorized into nine categories (i.e., normal (N), atrial fibrillation (AF), first-degree atrioventricular block (I-AVB), left bundle branch block (LBBB), right bundle branch block (RBBB), premature atrial contraction (PAC), premature ventricular contraction (PVC), ST-segment depression (STD), and ST-segment elevation (STE)). Some signals existed that contained two or three labels, and all signals with multiple labels were removed in this study.

### 2.2. Method Outline

The proposed method consists of three modules: data processing, visualized lead selection, and a ResBiTime network. During data processing, first, noise is removed from the original 12-lead ECG signals by a discrete wavelet transform algorithm, and the clean 12-lead signals are extracted heartbeats, by heartbeat detection. In addition, the oversampling method was used to balance the training set. Next, all heartbeats from the 12-lead signals are fed into the visualized lead selection module, which identifies heartbeats from effective leads. Visualized lead selection involves a lead-wise network and a determining-valid-leads process. Finally, the ResBiTime network, which combines a Bi-LSTM with residual connections, classifies heartbeats from the selected leads into nine categories. Figure 1 shows the workflow of the proposed method.

### 2.3. Data Preprocessing

#### 2.3.1. Removing Noise

All ECG signals were sampled at a frequency of 500 Hz. To attenuate interference such as power lines, discrete wavelet transform (DWT) was used on the original ECG signals to remove noise. The db6 wavelet decomposed an ECG recording into a nine-level wavelet, and each level of the wavelet component represented signal components in different frequency bands. Components D1, D2, D3, and A9 were removed because there is little information in ECG signals above 45 Hz. Finally, the remaining components, including D4, D5, D6, D7, D8, and A8, were reconstructed to obtain clean data [16]. Wavelet denoising is detailed in [17].

#### 2.3.2. Heartbeats Detection

The Pan–Tompkins algorithm [18] was used to detect the peak of the R wave. A total of 125 sample points to the left and 199 sample points to the right of the R peak were selected, forming a segment of 325 samples in length [4]. All categories of beats were randomly divided into a training set and a test set; 80% of each category of heartbeats was randomly selected for the training set and 20% of each category of heartbeats was randomly selected for the test set.

#### 2.3.3. Data Balancing

Oversampling is a common data balance method, but it may suffer from overfitting. In order to minimize the number of duplicate samples, a few arrhythmia categories in the training set with size of less than 8000 (i.e., LBBB and STE) were duplicated to balance data for 9-class classification, and the other categories maintained the original amount of samples (i.e., N, AF, IAVB, RBBB, PAC, PAV, and STD). Table 1 presents the number of ECG signals and the number of heartbeats before and after data were balanced.

### 2.4. The Visualized Lead Selection Module

Figure 2 illustrates the overall architecture of the visualized lead selection module. This module consists of a lead-wise network and a determining-valid-leads process. The heartbeats of 12 leads are fed into the lead-wise network to generate feature maps that are input into the determining-valid-leads process to obtain a lead activation heatmap (LA heatmap) [19]. Finally, based on the LA heatmap, five valid leads are selected.

#### 2.4.1. The Lead-Wise Network

The lead-wise network is used to individually extract features within the heartbeats of each lead and form feature maps, which are used to generate the LA heatmap. The lead-wise network contains nine convolutional layers and four pooling layers. Each convolutional layer is followed by a ReLU function, and each ECG lead is convolved separately. Following passage through the nine convolutional layers, the resultant feature maps undergo averaging in a global average pooling layer (GAP) [20].

The designed lead-wise network includes nine 1-D convolution layers and four pooling layers. For each 1-D convolution layer, the *k*-th feature map of the *c*-th ECG lead is obtained by
(1)$$h_{c}^{k}=x_{c}\times W_{c}^{k},$$
where *x_c_* represents the *c*-th lead of the ECG sample, × represents the 1-D convolution, and *W_c_^k^* represents the corresponding parameter in the convolution template for the *c*-th. Table 2 presents the detailed structure of the lead-wise network.

#### 2.4.2. Determining Valid Leads

The determining-valid-leads process selects valid leads from 12 leads by generating the LA heatmap, which can reflect the activations of different leads in the 9-class classification. During this process, for a given 12-lead ECG heartbeat, let *h_c_^k^*(*t*) represent the activation of the *k*-th kernel in the *c*-th ECG lead at the last convolution layer, where *t* represents the time step. The result of the GAP is represented as follows:(2)$$H_{c}^{k}=\sum_{t}h_{c}^{k}k\left(t\right)$$

For a specific category *s*, *y_s_* was obtained by calculating the weighted sum of the GAP output, represented as follows:(3)$$y_{s}=\sum_{c}\sum_{k}w_{c}^{k,s}H_{c}^{k}=\sum_{c}A_{c}^{s}$$
where *w_c_^k^*^,*s*^ represents the weight corresponding to the *k*-th kernel in the *c*-th ECG lead for category *s* in the linear layer. *A_c_^s^* represents the weighted sum of all feature maps for the *c*-th ECG lead, which is as follows:(4)$$A_{c}^{s}=\sum_{k}w_{c}^{k,s}H_{c}^{k}$$

The higher the *A_c_^s^*, the higher the probability for category *s*. Thus, *A_c_^s^* indicates the importance of the *c*-th ECG lead for category s. All *A_c_^s^* are normalized row-wise to obtain the lead activation heatmap. As the proposed lead-wise network only uses intra-lead convolution operations, all leads are processed separately, allowing the LA heatmap to reflect the activations of different leads.

### 2.5. The ResBiTime Network

The ResBiTime network was used for the 9-class classification of the selected 5-lead ECG heartbeats. This network merges the advantages of Bi-LSTM with residual connections to capture long-term and short-term dependencies in time series data, as well as complementary information from different leads.

Figure 3 shows the structure of the ResBiTime network. The ResBiTime network first processes input data using a Bi-LSTM layer to capture forward and backward temporal information. The network then progresses through a series of residual blocks, each comprising two parallel LSTM layers. One layer learns directly from the previous layer’s output, while the other learns from the original input. The outputs of these layers are passed on to the next layer. Between each LSTM layer, residual connections are introduced to allow information to pass directly from lower to higher layers. This design helps mitigate the common problem of vanishing gradients in deep networks and improves learning efficiency. The output of the LSTM layers is integrated and processed through a series of fully connected layers, which use the ReLU activation function and dropout to prevent overfitting. The ResBiTime network combines Bi-LSTM and residual connections, making the network excel in processing sequence data with complex temporal dependencies. Additionally, its multi-layered and bidirectional architecture enables it to capture deep features more effectively in time series and complementary features among leads.

## 3. Results

### 3.1. Evaluation Metrics

Three metrics, i.e., precision, recall, and F1-score, were used to evaluate the performance of the proposed method, and they are defined as follows:(5)$$Recall=\frac{tp}{tp+fn}$$
(6)$$Precision=\frac{tp}{tp+fp}$$
(7)$$F1=\frac{2\times Precision\times Recall}{Precision+Recall}$$
where true positive (*tp*) is defined as the number of heartbeats correctly classified as a given category. False negative (*fn*) is defined as the number of heartbeats that are in a category but incorrectly classified as another category. True negative (*tn*) is defined as the number of heartbeats that are in other categories and not classified as the given category. False positive (*fp*) is defined as the number of heartbeats that are in other categories but incorrectly classified as the given category.

### 3.2. Results

Experiments were performed with a NVIDIA A100 Tensor Core GPU (Santa Clara, CA, USA) and the Windows 10 operating system. PyTorch 1.4.0 was employed for the deep learning framework. The Kaiming initializer and an orthogonal initializer have been widely used to optimize parameters of a network [21]. Similarly, in this study, the Kaiming initializer [22] was used to initialize parameters of the lead-wise network, and the orthogonal initializer [23] was used to initialize parameters of the ResBiTime network, which were shown to greatly improve the converging speed of model parameters. In addition, the Adam optimizer [24] built into PyTorch was used to train both networks, which may have greatly reduced the influence of data balance. The lead-wise network was developed with the following hyperparameters: batch size, 32; 60 epochs; and a learning rate of 0.0005. The ResBiTime network was developed with the following hyperparameters: batch size, 32; 30 epochs; and a learning rate of 0.001. The proposed method utilized 5-fold cross-validation.

Figure 4 illustrates the average values of the LA heatmap across all samples. In the heatmap, yellow signifies greater activation, whereas blue denotes lesser activation. As shown in the LA heatmap, leads aVF, V1, V2, V3, and V6 exhibited higher activation in all categories. Leads aVF, V1, V2, V3, and V6 were consequently selected and fed into the ResBiTime network for classification. This approach effectively eliminated redundant leads in the 12-lead ECG heartbeats used for arrhythmia classification.

Currently, the gold standard for arrhythmia classification is mainly based on 12-lead electrocardiograms (ECGs) and diagnoses made by cardiologists. In this study, the results of expert annotations were used as the final validation. Table 3 shows precision and recall percentages and F1-scores of the proposed method for the test set. The best results from the 5-fold cross-validation were selected for presentation. The proposed method achieved the worst precision (88.88%), recall (86.09%), and F1-score (0.8764) in the PAC category. Conversely, in the LBBB category, the method recorded the highest precision at 97.32%, the highest recall at 96.11%, and an F1-score of 0.9671. The average values of the proposed method for precision, recall, and F1-score were 93.25%, 93.03%, and 0.9313, respectively.

### 3.3. Comparison with Previous Works for the CPSC 2018 Db

Table 4 provides a detailed comparison of the proposed method with previous studies. In this study, results for the 9-class classification (precision at 93.25%, recall at 93.03%, and an F1-score of 0.9313) were the highest compared to results of previous studies using the same classification category. Jiang et al. [5] proposed a multi-model and multi-scale fusion network that extracts more comprehensive features from multimodal ECG data, including 1-D ECG time series and 2-D time-frequency graphs, using various dilated convolutional kernels. This approach captures the abnormal patterns of diseases, and they achieved an accuracy of 84.91%, a recall of 82.64%, and an F1-score of 0.8352 for the 9-class classification. Wang et al. [9] proposed a multi-scale fusion network that captures the abnormal patterns of diseases by extracting multi-scale features from 12-lead ECG signals, and their study achieved a precision of 83.80%, a recall of 82.20%, and an F1-score of 0.8280 for the 9-class classification. Geng et al. [25] proposed a multi-task deep neural network that achieved a precision of 85.20%, a recall of 80.00%, and an F1-score of 0.8270 for the 9-class classification.

Table 4 demonstrates that previous studies achieved good performance using the CPSC 2018 database when classifying arrhythmias. However, these studies utilized 12-lead ECG signals that included unnecessary information due to redundant leads. Furthermore, the networks used in these studies overlooked the temporal dependencies inherent in ECG signals. This study employed visualized lead selection to choose five leads containing valid, non-redundant information for arrhythmia classification. Subsequently, the ResBiTime network, designed to fuse features from different leads, effectively learned deep features without compromising shallow information. Its Bi-LSTM structure provided a more comprehensive understanding of the context and effectively captured long-term temporal dependencies.

### 3.4. Comparison with Classical Baselines for the CPSC 2018 DB

The classic VGG and ResNet have demonstrated satisfactory performance in arrhythmia classification [9,30]. Additionally, the 12-lead ECG is considered the gold standard in clinical practice [5,25]. Therefore, in this study, we used the 12-lead ECG to develop VGG and ResNet for comparison with the proposed method.

Table 5 provides detailed results for 9-class classification using the test set. The proposed method yielded the best precision (93.25%), recall (93.03%), and F1-score (0.9305). For the baselines, ResNet achieved higher values (i.e., precision, 86.52%; recall, 87.58%; and an F1-score of 0.8688) than VGG did.

## 4. Discussion

### 4.1. Ablation Experiment

To evaluate the effectiveness of the visualized lead selection, a comparison experiment between the proposed method and the ResBiTime network was designed in this study.

Table 6 presents the results of the 9-class classification using the test set. The results of the ResBiTime network yielded 92.85% precision, 93.13% recall, and an F1-score of 0.9302. The memory usage of the ResBiTime network, at 674.52 MB, was significantly higher than the 396.61 MB required by the proposed method. This suggests that the visualized lead selection module effectively removed redundant leads in 12-lead ECG heartbeats for arrhythmia classification, thereby reducing the overhead of the ResBiTime network.

### 4.2. Interpretability Analysis

In the medical field, the interpretability of deep learning methods is crucial [32]. The divided lead convolution design of the lead-wise network enables the LA heatmap to provide interpretability from the perspective of individual ECG leads. Figure 4 shows the LA heatmap for all normal and arrhythmia samples. The yellow color represents greater activation, while the blue color represents lesser activation. In the LA heatmap, a larger value indicates that the lead has a high probability of the given arrhythmia class. Namely, the model infers a high probability of the given arrhythmia class. Combined with the activation information from all leads, the model can accurately judge the arrhythmia class. In the detection of N, I-AVB, and STD categories, the V6 lead showed the highest activation. In the detection of AF, LBBB, and RBBB categories, the V1 lead was the most activated. In the detection of the PAC category, V1, V2, and V6 leads were the most activated. In the detection of the PVC category, V2 and V6 leads were the most activated. In the detection of the STE category, V3 and V6 leads were the most activated. In the detection of all nine categories, aVF, V1, V2, V3, and V6 leads were primarily activated. Figure 5 shows the LA heatmaps for an individual sample. For instance, in the LA heatmap of sample A0001, categorized as RBBB, the V1 lead showed the most activation.

An ECG signals the curve of the heart’s electrical activity, and waveforms recorded by different leads can effectively reflect changes in various parts of the heart [33]. Bruce et al. [34] proposed that leads V1, V2, and V3 monitor the activity of the heart from the anterior surface; leads V1 and V2 examine the interventricular septum and right ventricular electrical activity; lead V3 monitors the activity of the left ventricular anterior wall; lead V6 measures the activity of the lower anterolateral wall of the left ventricle; and lead aVF observes the electrical activity on the coronal plane. Leads aVF, V1, V2, V3, and V6 provide different perspectives on the electrical activity of various parts of the heart. By combining the activated leads in each classification category depicted in the LA heatmap, we can analyze and identify the type of cardiac episode and changes in different heart areas from the neural network’s perspective.

## 5. Conclusions

This study proposed a method to classify nine categories of arrhythmia that achieved a precision of 93.25%, a recall of 93.03%, and an F1-score of 0.9313 using the CPSC 2018 database. The visualized lead selection of this method identified valid leads from 12-lead ECG signals, effectively reducing the memory overhead for network training, making it suitable for wearable devices. The proposed method used DWT to remove noise from ECG signals to accommodate data from different sources. To validate the effectiveness of this method, we employed 5-fold cross-validation and used evaluation metrics such as precision, recall, and F1-score. The oversampling method was used to balanced data. Through these measures, the proposed method can adapt to different datasets. However, the proposed method identified leads significant in arrhythmia classification, but failed to identify the specific waveforms within a heartbeat that are crucial for this classification. Future work may overcome this limitation by employing joint optimization and Grad-CAM [35] in the lead-wise network. Additionally, future research should focus on integrating the LA heatmap with dynamic physiological heart changes for a more comprehensive interpretation.

## Figures and Tables

**Figure 1 bioengineering-11-00578-f001:**
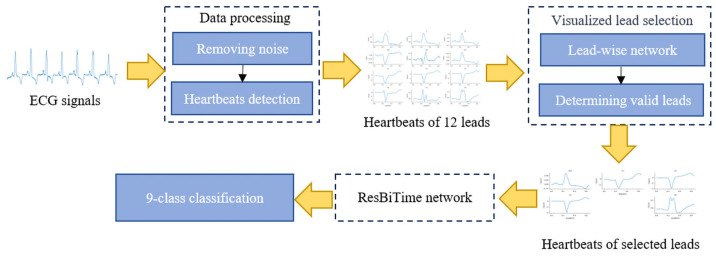
Workflow of the proposed method.

**Figure 2 bioengineering-11-00578-f002:**
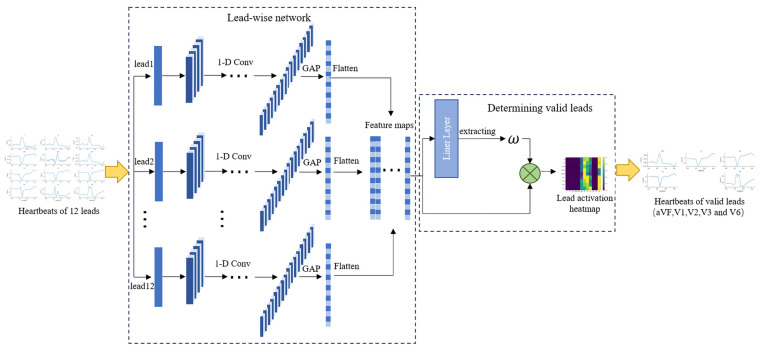
The architecture of the visualized lead selection module. GAP: global average pooling layer. Flatten: flatten multi-dimensional inputs into a 1-D format.

**Figure 3 bioengineering-11-00578-f003:**
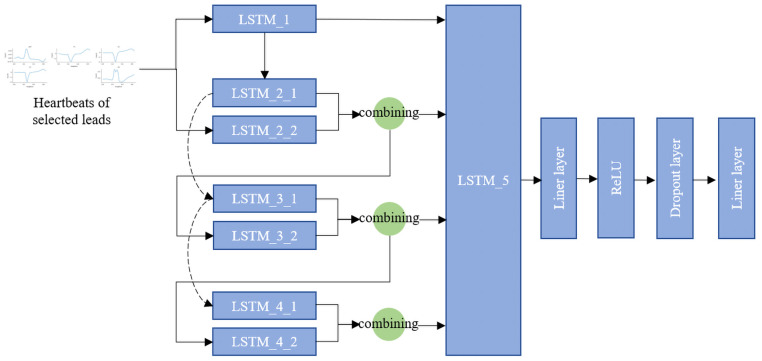
The architecture of the ResBiTime network. Combining: combining the input features. The dashed lines in the diagram represent residual connections.

**Figure 4 bioengineering-11-00578-f004:**
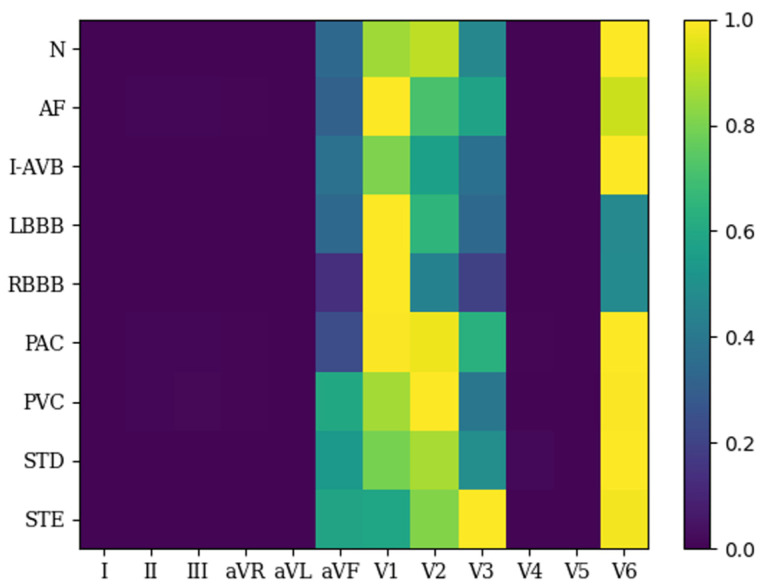
Lead activation heatmap for all samples.

**Figure 5 bioengineering-11-00578-f005:**
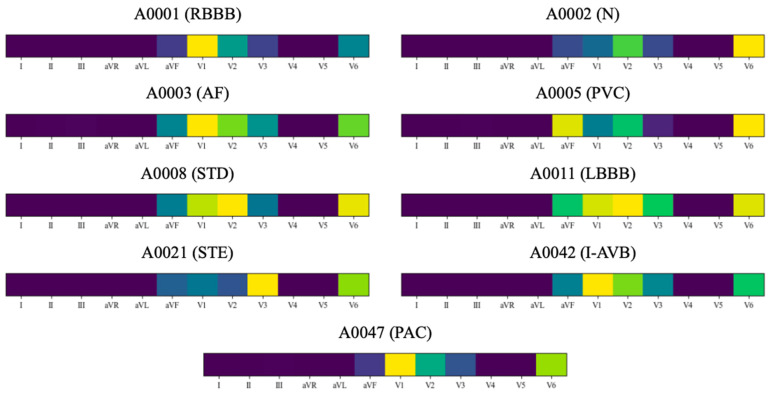
Lead activation heatmaps for an individual sample. Yellow indicates greater activation, while blue indicates lesser activation.

**Table 1 bioengineering-11-00578-t001:** The number of heartbeats in each category.

Categories	Number of Signals	Number of Heartbeats				
		Training Set		Test Set	Total Number	
		Before	After		Before	After
N	918	13,027	13,027	3314	16,341	16,341
AF	1098	12,784	12,784	3168	15,952	15,925
IAVB	704	8248	8248	2159	10,407	10,407
LBBB	207	2356	7068	566	2922	7634
RBBB	1695	19,512	19,512	4959	24,471	24,471
PAC	556	9201	9201	2293	11,494	11,494
PAV	672	11,491	11,491	2842	14,333	14,333
STD	825	10,939	10,939	2653	13,592	13,592
STE	202	2733	8199	619	3352	8818
Total	6877	90,291	100,469	22,573	112,864	123,069

**Table 2 bioengineering-11-00578-t002:** The detailed structure of the lead-wise network.

Layer Name	Output Size	Kernel Size/Stride
Input	(12, 1, 325)	-
Cov1	(12, 12, 325)	2/1
Cov2	(12, 12, 325)	1/1
Cov3	(12, 24, 325)	24/2
Maxpooling	(12, 24, 163)	3/1
Cov4	(12, 24, 163)	1/1
Cov5	(12, 32, 163)	32/1
Maxpooling	(12, 32, 163)	3/1
Cov6	(12, 32, 163)	1/1
Cov7	(12, 48, 163)	32/1
Maxpooling	(12, 48, 163)	3/1
Cov8	(12, 48, 163)	1/1
Cov9	(12, 60, 163)	48/1
Maxpooling	(12, 60, 163)	3/1
GAP	(12, 60)	-
Linear	9	-

**Table 3 bioengineering-11-00578-t003:** The 9-class classification results of the proposed method for the test set.

True Labels	Predicted									Precision (%)	Recall (%)	F1-Score
	N	AF	IAVB	LBBB	RBBB	PAC	PVC	STD	STE			
N	3189	9	9	0	18	30	26	26	7	95.79	96.23	0.9601
AF	20	2823	47	4	62	81	86	40	5	88.85	89.11	0.8883
IAVB	6	49	2043	0	12	14	24	9	2	93.50	94.63	0.9406
LBBB	0	9	1	544	1	1	6	4	0	97.32	96.11	0.9671
RBBB	7	53	10	1	4818	24	37	9	0	96.57	97.16	0.9686
PAC	34	119	35	1	21	1974	55	52	2	88.88	86.09	0.8746
PVC	47	96	25	9	35	56	2506	63	5	89.66	88.18	0.8891
STD	19	26	12	0	20	38	53	2485	0	92.34	93.67	0.9300
STE	7	4	3	0	2	3	2	3	595	96.59	96.12	0.9636
Average	-	-	-	-	-	-	-	-	-	93.25	93.03	0.9313

**Table 4 bioengineering-11-00578-t004:** Comparison of this work with previous works.

Authors	Year	Themes	Database	Methods	Precision (%)	Recall (%)	F1-Score
Chen et al. [26]	2020	9-class	The CPSC 2018 DB	CNN	-	-	0.8400
Wang et al. [9]	2020	9-class	The CPSC 2018 DB	Multi-scale CNN	83.80	82.20	0.8280
Liu et al. [27]	2021	9-class	The CPSC 2018 DB	NAS-TCAM-S	-	-	0.7813
Li et al. [6]	2022	9-class	The CPSC 2018 DB	CNN + Channel Attention + ensemble model	84.47	80.31	0.8170
Le et al. [28]	2022	9-class	The CPSC 2018 DB	X3ECG w/HC + DDI	-	-	0.8140
Zhang et al. [29]	2023	3-class	The CPSC 2018 DB	1-D CNN + Fine-tuning	-	-	0.8200
Jiang et al. [5]	2023	9-class	The CPSC 2018 DB	Multi-scale + Multi-model CNN	84.91	82.64	0.8352
Geng et al. [25]	2023	9-class	The CPSC 2018 DB	SE-ResNet + task-specific model	85.20	80.00	0.8270
This work	2024	9-class	The CPSC 2018 DB	Lead selection + ResBiTime	93.25	93.03	0.9313

**Table 5 bioengineering-11-00578-t005:** Comparison of this work with classical baselines.

Neural Network	Precision (%)	Recall (%)	F1-Score
VGG [17]	81.52	81.16	0.8108
ResNet [31]	86.52	87.58	0.8688
**This work** (i.e., lead-wise network + ResBiTime network)	**93.25**	**93.03**	**0.9313**

**Table 6 bioengineering-11-00578-t006:** Comparison of this work with the ResBiTime module.

Neural Network	Precision (%)	Recall (%)	F1-Score	Memory Usage (MB)
ResBiTime network	92.85	**93.13**	0.9302	674.52
**This work** (i.e., lead-wise network + ResBiTime network)	**93.25**	93.03	**0.9313**	**396.61**

## Data Availability

The dataset used in this study is a public dataset, and it can be downloaded from http://2018.icbeb.org/Challenge.html, accessed on 23 November 2023.

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
