# Peer review of "Visualized Lead Selection for Arrhythmia Classification Based on a Lead Activation Heatmap Using Multi-Lead ECGs"

_bioengineering, 2024, doi:10.3390/bioengineering11060578_

Round 1

Reviewer 1 Report

Comments and Suggestions for Authors

The manuscript discusses a visualized lead selection method for classifying arrhythmias in multi-lead ECG signals. This topic is of interest to the journal and some comments are provided for improvements:

1.      The manuscript lacks some technical details, making it difficult for readers to follow. For example, the descriptions of the learning rates, batch sizes, and epochs for the networks should be clearer. Please provide additional clarification.

2.      The rationale for using specific methods, such as the Adam optimizer and visualized lead selection, needs a more thorough explanation.

3.      Please elaborate on how the method might be adapted or validated on different datasets, and how it could manage varying levels of data quality, as this would offer valuable insights for readers.

Comments on the Quality of English Language

Minor changes.

Reviewer 2 Report

Comments and Suggestions for Authors

1. The text mentions 'Discrete Wavelet Transform (DWT) was used on the original ECG signals to remove noise. The components D1, D2, D3, and A9 were removed because there is very little information in ECG signals above 45 Hz.' Since DWT is an applied technology in this context, it should briefly explain its methodology and function. Moreover, it does not clearly specify which components were extracted.

2. The text mentions, 'Table 1 presents the number of ECG signals and the number of heartbeats extracted for 125 categories.' It can be seen that the number of samples for each category is different. Will the imbalance or balance in sample sizes affect the accuracy?

3. In Equation (1), Xc has 'c' as a subscript, but in the text, it is not a subscript. Please ensure that all parameters in the equations are consistently formatted.

4.This paper should explain the innovations introduced in this study.

5.The software and hardware environment for algorithm execution should be explained in the results section.

Reviewer 3 Report

Comments and Suggestions for Authors

This paper presented visualization of ECG lead selection for arrhythmia classification. The concept itself is interesting, but the authors did not compare the proposed method with gold standard. It is therefore recommended with major revision.

1. The authors claimed visualization of lead selection. What is the current gold standard of arrhythmia classification? How many leads do the clinics use, and what guidance is used for classification? It is suggested for deep discussion as the paper's benchmark.

2. The authors used LSTM with ResNet for classification of heartbeat categories. In table 1, the dataset for different categories are highly biased and imbalanced. How did the authors treat the biased dataset? Can LSTM generalize to the small category of STE? It is suggested to add metrics for each category.

Comments on the Quality of English Language

It is suggested to polish the words in the abstract and introduction, and make the background and motivation of this study.

Round 2

Reviewer 2 Report

Comments and Suggestions for Authors

Authors revised the draft according the reviewer's comments.

Reviewer 3 Report

Comments and Suggestions for Authors

The authors have improved their manuscript and addressed my concerns. It is recommended for publication.